The gut microbiota in mice with erythropoietin—induced abdominal aortic aneurysm

Lyu Xinyi 1
Jiang Mingjun 1
Shi Jiahao 1
Liu Qi 1
Liu Xilian 2
Li Yulan 2
Ding Shu-Qin 3
Dai Xianpeng 2003020003@usc.edu.cn 1
1 Department of Vascular Surgery, The Second Affiliated Hospital, Hengyang Medical School, University of South China , Hengyang , Hunan , China
2 Department of Endocrinology and Metabolism, The Second Affiliated Hospital, Hengyang Medical School, University of South China , Hengyang , Hunan , China
3 Clinical Laboratory, The First Affiliated Hospital of Bengbu Medical University, Anhui University , Bengbu , Anhui , China
Gelfand Mikhail
Electronic publication date: 2025 Apr 7
Publication date: 2025
Volume: 13
Electronic Location ID: e19222
Received 2024 Nov 22; Accepted 2025 Mar 6
Copyright: ©2025 Lyu et al.
Copyright year: 2025
Copyright holder: Lyu et al.
License: This is an open access article distributed under the terms of the Creative Commons Attribution License, which permits unrestricted use, distribution, reproduction and adaptation in any medium and for any purpose provided that it is properly attributed. For attribution, the original author(s), title, publication source (PeerJ) and either DOI or URL of the article must be cited.
License URL: https://creativecommons.org/licenses/by/4.0/

Keywords: Abdominal aortic aneurysm, Erythropoietin, Gut microbiota, 16S rRNA-sequencing

Funding: Natural Science Foundation of Hunan Province 2022JJ70033 Postgraduate Scientific Research Innovation Project of Hunan Province CX20240849 This work was supported by Natural Science Foundation of Hunan Province (2022JJ70033), and Postgraduate Scientific Research Innovation Project of Hunan Province (CX20240849). The funders had no role in study design, data collection and analysis, decision to publish, or preparation of the manuscript.

==============================
Background

In recent years, a novel animal abdominal aortic aneurysm (AAA) model was established by administering erythropoietin (EPO) to wild-type (WT) mice. However, the influence of EPO on the murine fecal microbiota remains uninvestigated. Therefore, this study aims to explore the potential association between gut microbiota changes and AAA development in this model.

Methods and results

Adult male C57BL/6 mice were used to establish the AAA model by intraperitoneal administration of recombinant human EPO at a dosage of 10,000 IU/kg daily for 28 consecutive days. Hematoxylin and eosin (H&E) and Elastin Van Gieson (EVG) staining revealed that EPO administration increased aortic wall thickness and diameter, accompanied by enhanced degradation of the elastic lamina. The 16S rRNA—sequencing data were deposited in the Sequence Read Archive (PRJNA1172300). LEfSe analysis revealed that Akkermansia, Lawsonibacter, Clostridium, and Neglectibacter were significantly associated with EPO-induced AAA development, while Lactobacillus, Alistipes, Limosilactobacillus, and Eisenbergiella showed significant negative correlations. Analysis using the Kyoto Encyclopedia of Genes and Genomes (KEGG) prediction module revealed significant differences in metabolic pathways between the two groups, including alanine, aspartate and glutamate metabolism; cysteine and methionine metabolism; pyrimidine metabolism; carbon metabolism; ABC transporters; and oxidative phosphorylation pathways.

Conclusions

EPO-induced gut dysbiosis, particularly changes in Akkermansia, Lactobacillus, and Alistipes abundance, may contribute to AAA formation via inflammation, oxidative stress, and metabolic dysfunction. While this model advances AAA research, its limitations underscore the need for human validation and mechanistic studies. Future work should prioritize multi-omics integration and cross-model comparisons to unravel the complex microbiota-AAA axis.

Introduction

Abdominal aortic aneurysm (AAA) is a significant public health concern, characterized by the localized dilatation of the abdominal aorta, with a diameter at least 1.5 times its normal size (Musto et al., 2024). This critical medical condition has the potential to precipitate rupture, which may result in acute exsanguination and is associated with elevated mortality rates in the absence of therapeutic intervention (Troisi et al., 2023). Despite the progress in medical and surgical interventions, the management of AAA remains challenging, primarily due to an incomplete understanding of its etiology and the absence of universally effective medical treatments (Troisi et al., 2023).

The angiotensin-II (AngII) perfusion is the most commonly used AAA animal model, but it requires ApoE gene knockout (ApoE−/−) mice (Golledge, Lu & Curci, 2024; Wu et al., 2024; Xu et al., 2024). A novel AAA model induced by erythropoietin (EPO) has recently been developed in wild-type (WT) mice, producing outcomes comparable to those observed in ApoE−/− mice (Zhang et al., 2024; Zhang et al., 2021; Zhang et al., 2023). The primary mechanism underlying EPO-induced AAA in murine models involves angiogenesis and inflammatory cell infiltration (Zhang et al., 2021). The analysis reveals distinct pathogenic mechanisms underlying the two models, which warrant further investigation.

The gut microbiome comprises a complex ecosystem of trillions of microorganisms, including bacteria, viruses, fungi, archaea, and eukaryotes, that inhabit the gastrointestinal tract. This intricate community plays a crucial role in maintaining human health (Buttar et al., 2024). In the eubiotic state, the gut microbiome supports the host’s mucosal immune defense and energy homeostasis, along with other physiological processes. Changes in the gut microbiome composition, leading to dysbiosis, may contribute to the pathogenesis of various diseases (Gao et al., 2018). Recent studies have suggested a potential link between changes in gut microbiota composition and the development of AAA in a murine model infused with Ang II (Xiao et al., 2023; Xie et al., 2020). However, the interaction between gut microbiota and the effects of EPO in the murine AAA model remains undocumented in the scientific literature. Therefore, this study aims to elucidate the mechanism by which EPO induces AAA development, with a particular focus on the role of gut microbiota.

Materials and Methods

AAA mice model of EPO injection

All experimental protocols involving animals were approved by the Animal Care Ethics Committee of Bengbu Medical University prior to initiation. The study protocol received ethical approval by the Animal Ethics Committee under the registration number 2024-061. Twenty healthy male C57BL/6 mice, aged 10 weeks and weighing between 30 and 32 grams, were obtained from Changzhou Cavens Laboratory Animal Ltd (Changzhou, China). The mice were housed in a controlled environment with a 12-hour light/dark cycle, ensuring adequate ventilation. Each cage housed four mice; provided unrestricted access to food and water. Ambient room temperature was maintained between 20 to 22 degrees Celsius, with relative humidity levels kept at 30% and 70%. Subjects were randomly assigned to two experimental groups using a random number table: the control group (n = 8) received normal saline, whereas the AAA group (n = 12) was administered intraperitoneal injections of recombinant human EPO, produced by 3SBio Inc. (Shenyang, China), at a dose of 10,000 IU/kg/day for four weeks to induce the AAA condition (Zhang et al., 2021).

Specimen collection

A power analysis was first calculated using the online RNASeqPower tool (https://rodrigo-arcoverde.shinyapps.io/rnaseq_power_calc/). The sample size (six controls, seven EPO-treated) was determined to achieve 83% statistical power (α = 0.05), which balances ethical considerations (reducing animal use) and resource constraints while ensuring robust detection of microbial community differences. Intestinal waste specimens were collected on the 28th day and stored at −80 °C for DNA isolation and 16S ribosomal RNA gene sequencing. After specimen collection, the rodents were euthanized via intraperitoneal injection of a mixture containing ketamine (80 mg/kg) and xylazine (10 mg/kg). The animals were then perfused with 10 mL of ice-cold 0.01 M phosphate-buffered saline (PBS, pH 7.4), followed by fixation with an equal volume of ice-cold 4% paraformaldehyde in PBS. After perfusion, the aortic segments were excised, with six samples collected from each experimental group. The segments were then fixed by immersion in 4% PFA in 0.01 M PBS (pH 7.4) overnight, followed by transfer to a 30% sucrose solution in PBS at 4 °C for an additional overnight period. Subsequently, the specimens were embedded in OCT mounting medium (Tissue-Tek, Miles, Elkart, IN, USA). Longitudinal sections of 5 µm thickness were cut using (model Leica CM1900, Bannockburn, IL, USA), and mounted on poly-L-lysine-coated slides to ensure optimal adhesion after thawing.

Histological analyses

The aortic tissue was stained using the Hematoxylin and Eosin (H&E) Staining Kit (Beyotime Biotechnology, Shanghai, China) to visualize vascular morphology. Subsequently, the Collagen Fiber and Elastic Fiber Staining Kit (EVG-Victoria Method, Solarbio, Beijing, China) was used according to the manufacturer’s protocol, to evaluate the degradation of the medial elastic lamina.

DNA extraction and 16S rRNA-sequencing

Bacterial genomic DNA extraction, primer adapter design and synthesis, polymerase chain reaction (PCR) amplification, product purification, quantification, and normalization of PCR products, as well as library preparation and sequencing, were performed by Majorbio (Shanghai, China). Approximately 200 mg of stool sample was meticulously collected from each mouse, and genomic DNA was subsequently isolated using the QIAamp DNA Stool Mini Kit (Qiagen, Hilden, Germany), according to the manufacturer’s protocol. Following the extraction of genomic DNA, the integrity and concentration of the DNA were assessed using 1.0% agarose gel electrophoresis and the NanoDrop® ND-2000 spectrophotometer (Thermo Fisher Scientific Inc., Waltham, MA, USA). Bacterial 16S rRNA genes were amplified via polymerase chain reaction (PCR) using the GeneAmp® PCR System 9700 (Applied Biosystems, Foster City, CA, USA). The forward primer, designated as 27F, and the nucleotide sequence 5′-AGRGTTYGATYMTGGCTCAG-3′, while the reverse primer, identified as 1492R, and the sequence 5′-RGYTACCTTGTTACGACTT-3′. Three replicates per sample were performed under standardized experimental conditions. DNA fragments were excised from a 2% agarose gel and subsequently purified in accordance with the protocol provided by the manufacturer, using the AxyPrep DNA Gel Extraction kit (Axygen Biosciences, Union City, CA, USA). Quantification was performed using the Quantus™ Fluorometer (Promega Corporation, Madison, WI, USA). Subsequently, the samples were diluted in proportion to the sequencing volume required for each individual sample. The TruSeqTM DNA Sample Preparation Kit (Illumina, San Diego, CA, USA) was employed for the construction of the DNA library in accordance with the manufacturer’s protocol. The raw data were acquired by sequencing the samples using the Illumina NovaSeq6000 platform.

Microbial population analysis

Quantitative Insights into Microbial Ecology (QIIME-2, release 2022.11) was employed to conduct the comprehensive data analysis (Bolyen et al., 2019). The sequencing reads were processed using a clipping and de-noising protocol that utilized the DADA2 algorithm (Callahan et al., 2016). The alpha diversity index was employed to evaluate species richness, diversity, and additional ecological metrics within the community, using the mothur software package (version 1.30.2). The similarity among operational taxonomic units (OTUs) was assessed using an index analysis, with the evaluation threshold set at a 97% (0.97) similarity level. A Venn diagram was employed to determine the intersection and unique species counts at the ASV level. Subsequently, bar diagrams were constructed to illustrate the composition of community structures across various samples at discrete taxonomic ranks, statistical analyses and graphical mappings were facilitated by tools in the R language (version 3.3.1). Circos-0.67-7 (http://circos.ca/) was used to produce Circos maps of communities, reflecting the proportion of dominant species composition in each group and the distribution proportion of each dominant species in different groups. The Kyoto Encyclopedia of Genes and Genomes (KEGG) orthologs (KO), Enzyme Commission (EC) numbers, Clusters of Orthologous Groups (COG), and MetaCyc metabolic pathways were inferred using the Phylogenetic Investigation of Communities by Reconstruction of Unobserved States (PICRUSt2, version 2.2.0) algorithm.

Statistical analysis

Statistical analyses were conducted using IBM SPSS Statistics version 22.0 (SPSS Inc., Armonk, NY, USA) for computational processing and data assessment. The independent samples Student’s t-test was employed to assess the statistical significance of the differences between the two groups. The Bray-Curtis dissimilarity metric was employed for the quantification of Beta diversity. Subsequently, the dataset was subjected to principal coordinate analysis (PCoA) for graphical representation. The Gut Microbiota Health Index (GMHI) utilized the Wilcoxon rank sum test, incorporating the Benjamini–Hochberg procedure for false discovery rate (FDR) multiple testing correction, to evaluate the health status by analyzing the species-level taxonomic features of gut microbiome samples. Linear discriminant analysis effect size (LEfSe) was employed to evaluate the disparities between the two groups. A linear discriminant analysis (LDA) score exceeding 2.0, coupled with a p-value less than 0.05, was considered indicative of a statistically significant difference.

Results

EPO-induced mouse AAA model

In the EPO injection group, the mortality rate was 16.67%, with two out of twelve mice succumbing to the intervention. Among the surviving mice, eight were developed AAA. In contrast, no AAA cases were observed in the control group. Representative images of aortic specimens from the two distinct groups were shown in Fig. 1A. H&E staining demonstrated that the average diameter of the aorta in the control group (Fig. 1B) was 0.83 cm (0.83 ± 0.08 cm, n = 6), while in the AAA group (Fig. 1C), it was 2.03 cm (2.03 ± 0.26 cm, n = 7). Compared with the control, EPO treatment significantly increased vascular diameter (p < 0.01, Fig. 1D). EVG-Victoria staining analysis revealed that, compared to the control group (Fig. 1E), EPO administration significantly enhanced elastic layer degradation, as shown in Fig. 1F. These findings suggest that EPO can effectively induce AAA development in murine models with a high success rate.

Figure 1 Identification of the EPO-induced AAA mouse model.

(A) Photographic representations of aortic specimens from the control group (con) and the EPO- treated group (AAA). (B and C) Typical H&E staining micrographs depicting the abdominal aorta from the control group (B) and EPO- treated group (C). (D) The statistical graph of the vascular diameter. Data represent the mean ± SD (n = 6 and 7 in the control and AAA groups, respectively). **P < 0.01 (Student’s t-test). (E and F) Representative EVG staining pictures of the abdominal aorta of control (E) and EPO-treated mice (F).

The alpha and beta diversities of the gut microbiota

The 16S rRNA gene sequencing was used to examine the effect of EPO on gut microbiota composition. Figure 2A showed that the rarefaction curve effectively demonstrates the sequencing dataset’s logical integrity and reflected the species richness within the sample. As the sequence count increased, the curves for all sample datasets became smoother, indicating that the sequencing depth was sufficient for subsequent bioinformatics analyses.

Figure 2 The alpha and beta diversities of the gut microbiota.

(A) Rarefaction curve. (B) ACE index. ** p < 0.01, Student’s t-test (C) Shannon diversity index. p > 0.05, Student’s t-test (D and E) 2D (D) and 3D (E) PCoA analysis. con: control group; AAA: EPO-treated group.

Alpha diversity analysis was used to assess the taxonomic richness and evenness of the gut microbiota community. The ACE and Shannon indices were applied to quantify species richness and evaluate sequencing depth adequacy. Analysis of the ACE index revealed a statistically significant increase in the EPO-treated group compared to the control group (Fig. 2B, P < 0.05). In contrast, the Shannon index showed no significant difference between the control and EPO groups (Fig. 2C, P > 0.05), indicating that while community diversity was similar between the two groups, the EPO group exhibited greater richness.

Beta diversity, which measures differences in species abundance distributions between the two groups, was analyzed using both two-dimensional (2D) and three-dimensional (3D) PCoA. The analysis showed that the intestinal microbiota of the control and EPO groups formed distinct clusters, indicating a significant difference in gut flora composition between the two groups (Fig. 3C and 3D, P < 0.05).

Figure 3 Analysis of the GMHI.

(A) Analysis of the difference of GMHI between control (con) and EPO (AAA) groups; (B) GMHI stratified control (con, n = 6) and EPO (AAA, n = 7) groups more strongly than Chao diversity. Each point in the scatter plot corresponds to a metagenomic sample (12 in total). The histogram shows the distribution of control (con, blue) and EPO (AAA, orange) samples based on the parameters of each axis.

EPO decreased the gut microbiome health index

GMHI is a metric for assessing individual health status based on the species-level taxonomic profiling of gut microbiome samples, which aids in the diagnosis of various diseases. Figure 4A showed a statistically significant reduction in the GMHI in the EPO-treated mouse group compared to the control group, with a p-value of 0.003405 indicating strong significance. The GMHI and Shannon diversity metrics for each sample were compared to evaluate consistency across the sample population. Figure 4B showed that the GMHI more effectively distinguishes the control and EPO groups, with a correlation coefficient (corr) of −0.56 and a p-value of 0.04, compared to the Shannon diversity index. The data further support the hypothesis that EPO treatment reduces the gut microbiome health index.

Figure 4 Changes in gut microbiota between the control group and the EPO-treated group.

(A) Venn diagram of ASV. (B and C) Bar diagram at phylum level (B) and genus level (C). (D) Circos diagram at phylum level (left) and genus level (right). con: control group; AAA: EPO-treated group.

EPO changed the gut microbiota

Analysis of the Venn diagram (Fig. 4A) demonstrated alterations in the composition of the microbiota within the EPO group. A total of 1,348 ASVs were identified in the control and EPO groups, with 539 unique ASVs in the EPO group and 418 in the control group. Notably, 391 ASVs were shared between the two groups (Fig. 4A). Bioinformatic analyses utilizing QIIME2 revealed the intestinal microbiota composition at the phylum level in both groups. The control group was dominated by Firmicutes, Bacteroidota, and Proteobacteria, representing over 90% of the microbiome. In contrast, the EPO group primarily composed of Firmicutes, Verrucomicrobia, Bacteroidota, and Proteobacteria (Fig. 4B and 4D). Bacteria with a relative abundance >5% at the genus level were selected for further analysis. Mice treated with EPO showed a significant increase in the relative abundance of unclassified Oscillospiraceae, Akkermansia, Ligilactobacillus, and Lawsonibacter compared to the control group. At the same time, the relative abundance of Lactobacillus and unclassified Lachnospiraceae decreased significantly (Fig. 4C and 4D). These findings suggest that EPO can modulate the intestinal microbiota composition in mice.

Species differences and marker species analysis

A comparative analysis between the two groups revealed genus-level differences (Fig. 5). The abundance of Akkermansia, Lawsonibacter, Clostridium, and Neglectibacter significantly increased in the EPO group compared to the control group (p < 0.05, Fig. 5A). The abundance of Lactobacillus, Alistipes, Limosilactobacillus, and Eisenbergiella significantly decreased (p < 0.05, Fig. 5A).

Figure 5 Analysis of species differences.

(A) The two groups were compared by Student’s t-test. The red bar was the control group (con) and the blue bar was the EPO-treated group (AAA); (B, C) Linear discriminant analysis (LDA) effect size (LEfSe) was used to analyze, the differences between the control group (con) and the EPO-treated group (AAA) was shown by cladogram (B) and histogram (C, at genus level). The red bar was the control group and the blue bar was the EPO-treated group, LDA > 2.

LefSe was used to identify differentially abundant bacterial taxa between the two groups, focusing on those with statistically significant differences. The LDA threshold was set at 2. The relationships among taxa from phylum to species level are shown in Fig. 5B. At the genus level, Akkermansia, unclassified Oscillospiraceae family, Lawsonibacter, Clostridium, Faecalibaculum, Helicobacter, unclassified Eubacteriales, Mailhella, Ruminococcus, and Neglectibacter were the predominant taxa in the EPO treatment group (Fig. 5C; p < 0.05).

PICRUSt2 analysis to predict the potential role of the gut microbiota in AAA formation

PICRUST2 can predict clusters of orthologous groups (COGs) and metabolic pathways, such as those in the KEGG and MetaCyc, in the context of disease progression. Figures 6A and 6B showed significant variations in COG functional abundance for intracellular trafficking, secretion, and vesicular transport; inorganic ion transport and metabolism; and amino acid transport and metabolism (p < 0.05). KEGG pathway analysis via PICRUSt2 revealed significant variations in numerous pathways at the Pathway Level 3. Bacterial functionality was primarily linked to metabolic pathways and biosynthesis of secondary metabolites (Fig. 6C). Analysis of the top 30 pathways revealed significant differences in several key metabolic processes, including alanine, aspartate, and glutamate metabolism; cysteine and methionine metabolism; pyrimidine metabolism; carbon metabolism; ABC transporters; and oxidative phosphorylation (Fig. 6D, adjusted p < 0.05).

Figure 6 PICRUSt2 analysis predicted the potential role of the gut microbiota.

(A) COG function classification. (B) Difference between the control group (con) and the EPO-treated group (AAA) in COG function. *p < 0.05, Student’s T test. U: intracellular trafficking, secretion, and vesicular transport; P: inorganic ion transport and metabolism; E: amino acid transport and metabolism. (C) Heatmap of KEGG pathway level 3. (D) Difference between the EPO and control groups in KEGG pathway level 3. ***p <  0.001, **p < 0.01, *p < 0.05, Student’s T test.

Discussion

Recently, Zhang et al. (2021) developed a novel animal model in which AAA develops after EPO administration in WT mice. The mechanism of EPO-induced AAA formation is not yet fully understood, but existing studies suggest that EPO is involved in the pathogenesis of AAA through multiple pathways (Zhang et al., 2024; Zhang et al., 2021). For example, it stimulates macrophage activation and the release of pro-inflammatory cytokines, which may trigger inflammation in the arterial wall and further promote AAA formation (Dinc, 2023; Hou et al., 2024; Zhang et al., 2024; Zhang et al., 2021). EPO may also alter the phenotype of vascular smooth muscle cells (VSMCs), inducing them to transition to a synthetic phenotype, leading to structural damage and destruction of the vascular wall (Zhang et al., 2024; Zhang et al., 2021). Additionally, EPO might influence the gut microbiota composition, thereby affecting the host’s immune response and metabolic functions (Ling et al., 2023; Zhu et al., 2022). However, the potential mechanisms by which EPO regulation of gut microbiota might impact changes in the structure and function of the abdominal aorta media remains unexplored. Here, we demonstrate that EPO administration induces AAA formation alongside significant alterations in gut microbial composition and metabolic pathways, providing new insights into microbiota-mediated vascular pathology.

Our findings reveal that EPO treatment disrupts gut microbiota homeostasis, marked by increased abundance of Akkermansia, Lawsonibacter, Clostridium, and Neglectibacter and decreased Lactobacillus, Alistipes, Limosilactobacillus, and Eisenbergiella. These shifts correlate with AAA development, as evidenced by aortic wall thickening, elastic lamina degradation, and systemic inflammation. Akkermansia, a gram-negative anaerobic bacterium, is a key component of the intestinal microbiota, may compromise intestinal barrier integrity, facilitating lipopolysaccharide (LPS) translocation and subsequent macrophage activation in the aortic wall (He et al., 2022; Zhang et al., 2019). Studies have shown that Akkermansia levels are reduced in the Ang II-induced AAA model (Wang et al., 2024; Xiao et al., 2023; Xie et al., 2020). However, our findings suggest a positive correlation between Akkermansia and EPO-induced AAA development in mice. Another bacterium, Alistipes, is also been found to be contrary to our findings in the Ang II-induced AAA model (Xiao et al., 2023). Alistipes is commonly found in gut microbiota and has been associated with both beneficial and detrimental health effects, depending on the context. Some studies suggest that Alistipes may play a protective role in certain conditions, such as reducing inflammation (Ma et al., 2024); others have linked it to diseases like colorectal cancer, depression, and cardiovascular disorders (Rahal et al., 2024; Won et al., 2024; Yu et al., 2025). In our report, a decreased Alistipes is found in EPO-induced AAA mice. These divergences may reflect distinct AAA mechanisms: Ang II primarily drives oxidative stress (Ajoolabady, Pratico & Ren, 2024), whereas EPO promotes angiogenesis and macrophage activation (De Luisi et al., 2013). Lactobacillus species, commonly found in the gastrointestinal tract, oral cavity, and urogenital tract, play a crucial role in maintaining microbial balance. These bacteria enhance digestive and support immune responses. Additionally, they can influence the progression of certain diseases (Chee, Chew & Than, 2020; Huang et al., 2022; Steiner & Lorentz, 2021). Some studies have shown a significant reduction in Lactobacillus populations within the Ang II-induced AAA model (Wang et al., 2024; Xiao et al., 2023; Xie et al., 2020). In this study, we confirmed that Lactobacillus was reduced in EPO induced AAA mice, suggesting that it may protect against AAA formation. They may share similar mechanisms as follows: reduced Lactobacillus levels could diminish anti-inflammatory short-chain fatty acid (SCFA) production (Luo et al., 2023), exacerbating vascular inflammation—a hallmark of AAA progression (Hou et al., 2024). These mechanisms align with prior studies linking gut dysbiosis to vascular remodeling but highlight EPO-specific pathways distinct from Ang II models.

In addition to Akkermansia, Lactobacillus, and Alistipes, discussed above, no other significantly altered microbiota have been reported to be directly associated with AAA. Their potential role in gut dysbiosis, metabolite production, and immune modulation suggests it may indirectly influence the disease’s development. Further research is needed to elucidate its specific contributions and explore whether targeting them or their metabolic pathways could offer therapeutic benefits for AAA prevention or treatment.

The statistical analysis of COG functional abundance revealed significant discrepancies in the categories of intracellular trafficking, secretion, and vesicular transport; inorganic ion transport and metabolism; and amino acid transport and metabolism. Intracellular trafficking, secretion, and vesicular transport are primary mechanisms for the exchange of substances, energy, and information within cells (Cui et al., 2022). The formation and transport of vesicles facilitate the organized movement of diverse biomolecules within cellular compartments, providing the physical foundation for the execution and regulation of a multitude of cellular physiological functions (Cui et al., 2022). Inorganic ion transport and metabolic pathways are integral to the maintenance of sodium, potassium, magnesium, calcium, iron, phosphate, and other inorganic salt absorption processes, supporting normal physiological function (Xu et al., 2021). Amino acid transport and metabolism involve transport channel activity, metabolic enzyme functions, and metabolite level modulation (Wu et al., 2022). These findings suggest that EPO significantly affects material and energy transport, inorganic salt absorption, and amino acid metabolism, which may contribute to EPO-associated aortic disease pathogenesis.

KEGG pathway analysis identified significant perturbations in metabolic pathways, including alanine, aspartate and glutamate metabolism; cysteine and methionine metabolism; pyrimidine metabolism; ABC transporters; and oxidative phosphorylation. Altered amino acid metabolism (e.g., glutamate) might impair VSMC contractility, accelerating medial degeneration (Gallina et al., 2021). Cysteine/methionine metabolism alterations might reduce glutathione synthesis, increasing oxidative stress in the aortic wall (Kong et al., 2022). ABC transporters may facilitate the efflux of pro-inflammatory molecules, exacerbating vascular inflammation (Liu et al., 2021). Oxidative phosphorylation dysregulation could impair ATP production in VSMCs, promoting apoptosis and extracellular matrix degradation (Phadwal et al., 2021). Therefore, dysregulation of these pathways may impair VSMC function through multiple routes: (1) Reduced glutathione synthesis due to altered cysteine/methionine metabolism elevates oxidative stress, promoting extracellular matrix degradation; (2) ATP depletion from disrupted oxidative phosphorylation compromises VSMC contractility; (3) Glutamate metabolism imbalances may exacerbate immune dysregulation, amplifying aortic inflammation. These findings suggest that EPO-induced microbiota changes exacerbate AAA via intertwined metabolic and inflammatory cascades.

Notably, this study provides the first comparative analysis of the gut microbiota in EPO-induced AAA mice. This model offers advantages, including addressing some limitations of traditional AAA models and providing a novel approach for further exploration. The complex role of intestinal flora warrants further investigation into the relationships between EPO, AAA, and gut microbiota.

The EPO model avoids genetic modification (e.g., ApoE−/−), uses WT mice, and mimics angiogenesis-driven AAA, complementing inflammation-centric models like Ang II. However, EPO’s erythropoietic effects may confound results, and its relevance to human AAA (often atherosclerosis-linked) requires further study. This model is preferable for investigating angiogenesis-microbiota crosstalk or comorbidities like chronic kidney disease where EPO is dysregulated. In addition, this study has also some other limitations, such as being conducted in a mouse model, and human samples are also required for further validation. Although the EPO-induced mouse model of AAA mimics human AAA pathology, differences between human and mouse gut microbiota must be considered. While our study highlights the role of gut microbiota in EPO-induced AAA, key differences between murine and human gut microbiota, such as dominant phyla (e.g., higher Bacteroidetes abundance in humans vs. Firmicutes in mice), metabolic capabilities, and immune interactions, may influence the translatability of our findings. Additionally, murine models lack environmental and genetic diversity inherent to human populations. These differences underscore the need for validation in human studies, particularly in AAA patients with altered EPO levels or renal comorbidities.

In summary, EPO-induced gut dysbiosis, particularly changes in Akkermansia, Lactobacillus, and Alistipes abundance, may contribute to AAA formation via inflammation, oxidative stress, and metabolic dysfunction. While this model advances AAA research, its limitations underscore the need for human validation and mechanistic studies. Future work should prioritize multi-omics integration and cross-model comparisons to unravel the complex microbiota-AAA axis.

Additional Information and Declarations

Competing Interests

Author Contributions

Animal Ethics

Data Availability

The authors declare there are no competing interests.

Xinyi Lyu performed the experiments, analyzed the data, prepared figures and/or tables, authored or reviewed drafts of the article, and approved the final draft.

Mingjun Jiang performed the experiments, prepared figures and/or tables, authored or reviewed drafts of the article, and approved the final draft.

Jiahao Shi analyzed the data, prepared figures and/or tables, and approved the final draft.

Qi Liu analyzed the data, prepared figures and/or tables, and approved the final draft.

Xilian Liu performed the experiments, prepared figures and/or tables, and approved the final draft.

Yulan Li analyzed the data, prepared figures and/or tables, and approved the final draft.

Shu-Qin Ding performed the experiments, authored or reviewed drafts of the article, and approved the final draft.

Xianpeng Dai conceived and designed the experiments, authored or reviewed drafts of the article, and approved the final draft.

The following information was supplied relating to ethical approvals (i.e., approving body and any reference numbers):

The Animal Care Ethics Committee of Bengbu Medical University (registration number 2024-061).

The following information was supplied regarding data availability:

The sequences are available at NCBI SRA: PRJNA1172300.

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
