# Peer review of "The gut microbiota in mice with erythropoietin—induced abdominal aortic aneurysm"

_PeerJ, doi:10.7717/peerj.19222_

## Round 0.1 · original submission · Major Revisions

The reviewers have a number of substantial concerns regarding the purpose and design of the study that need to be addressed. There are also editorial comments, in particular, the language requires correction.

·

Basic reporting

- The manuscript is generally well-written, but some sections contain complex sentence structures that could be simplified
- Lines 311–317: Overly long sentence; could be split
- Line 370: Ambiguous phrasing

Experimental design

The study addresses a novel and relevant research question, exploring EPO-induced dysbiosis and its role in AAA formation. The scope aligns well with the journal's objectives
The methodology is clear and sufficiently detailed to allow replication

Validity of the findings

- Statistical methods are sound and well-described.
- Elaborate on the reliability of EVG-Victoria staining in assessing elastic layer degradation.

Additional comments

- Expand on limitations regarding murine-human gut microbiota differences.
- Include more context on how identified pathways (e.g., oxidative phosphorylation) align with AAA pathogenesis

Reviewer 2 ·

Basic reporting

no comment

Experimental design

no comment

Validity of the findings

no comment

Additional comments

Comment 1
In the Methods section under "Specimen collection," I am not entirely clear on the purpose of the power analysis. Why did the authors use RNASeqPower to calculate the sample sizes for the control and AAA groups? It seems that the authors used 6 control group samples and 7 experimental group samples for DNA extraction and 16S rRNA sequencing. This appears to be the minimum sample size for a statistical power of 0.83, according to RNASeqPower. What rationale did the authors use to determine the sample size? Isn't a larger sample size typically more efficient in terms of statistical power?

Comment2
In Figure 1A, the gross image of the mouse heart and vascular tissue shows the apex facing right. Please clarify why this orientation was chosen during the photography. In terms of vascular dissection techniques, there is still residual tissue on the vascular surface, which may affect the reader's interpretation and judgment. Figure C and Figure D both show HE staining of the abdominal aorta, but the image sizes are inconsistent. It is recommended to unify the image sizes for better visual harmony and consistency. Similarly, Figure F and Figure G show EVG staining, with varying sizes, and it is suggested that the image sizes be kept consistent. Additionally, the placement of Figure D and Figure G seems to be designed for aesthetic purposes or other reasons, and it would be helpful for the authors to explain the rationale behind this arrangement. The font color in the figure legend is inconsistent—black font is used in other figures, but red font is used in Figure 1. It is recommended to standardize the font color to avoid distracting the reader's attention. Overall, maintaining consistency in image size, font color, and panel layout will enhance the visual appeal and clarity of the figure, making it more attractive and easier to read.

Comment 3
The mechanism of EPO-induced AAA formation is not yet fully understood, but existing studies suggest that EPO is involved in the pathogenesis of AAA through multiple pathways. For example, it stimulates macrophage activation and the release of pro-inflammatory cytokines, which may trigger inflammation in the arterial wall and further promote AAA formation. EPO may also alter the phenotype of vascular smooth muscle cells, inducing them to transition to a synthetic phenotype, leading to structural damage and destruction of the vascular wall. Additionally, EPO could influence the gut microbiota composition, thereby affecting the host’s immune response and metabolic functions. The dysbiosis induced by EPO may exacerbate inflammation and metabolic dysfunction, providing a favorable condition for the development of AAA. This study focuses on the mediating role of gut microbiota, which could offer new insights for the treatment and prevention of AAA. However, in the discussion section, we hope to include potential mechanisms by which EPO regulation of gut microbiota might impact changes in the structure and function of the abdominal aorta media.

Reviewer 3 ·

Basic reporting

Please discuss the differences in the flora found in this study and those previously reported in AAA patients or other AAA animal models.

The pros and cons of the EPO model relative to other common AAA models need to be added to the Discussion section. Under what circumstances is the EPO-induced AAA model preferred?

The figures in the manuscript need to be re-optimized and formatted.

Experimental design

There are false positives in the omics experiments, and the sample size of this study is tiny. The significant flora found in this study needs to be validated in new mouse or patient samples to improve the reliability of the results.

Validity of the findings

How many centimeters was the average diameter of the aorta in the disease group and the control group? Please explain in the Result section.

The potential mechanism by which EPO promotes AAA formation and the mechanism by which EPO may affect the relevant flora and pathways need to be discussed.

Additional comments

The email of the corresponding author needs to be replaced with an institutional email.

Please replace “Cohort” with “Group” in the manuscript.

What does' the distal and central aortic aneurysm 'mean?

Annotated reviews are not available for download in order to protect the identity of reviewers who chose to remain anonymous.

---

## Round 0.2 · accepted · Accept

The reviewers and I are satisfied with the corrections.


Reviewer 2 ·

Basic reporting

no comment

Experimental design

no commentno comment

Validity of the findings

no comment

Additional comments

no comment